# Leveraging Social Determinants of Health (SDoH) Knowledge Graph to Identify Latent Patterns in Veteran Suicide Risk

Chuming Chen*
*Wilmington VA Medical Center*
Chuming.Chen@va.gov

Fahmida Liza Piya*
*University of Delaware*
lizapiya@udel.edu

Joshua A. Rolnick
*Philadelphia VA Medical Center*
Joshua.Rolnick@va.gov

Suzanne A. Milbourne
*Wilmington VA Medical Center*
Suzanne.Milbourne@va.gov

Cathy H. Wu
*University of Delaware*
wuc@udel.edu

Thomas M. Powers
*University of Delaware*
tpowers@udel.edu

Jonathan Sanchez Garcia
*MDClone*
jonathan.sanchez-garcia@mdclone.com

Vinod Aggarwal
*MDClone*
vinod.aggarwal@mdclone.com

Aidong Zhang
*University of Virginia*
aidong@virginia.edu

Rahmatollah Beheshti
*University of Delaware*
rbi@udel.edu

## ABSTRACT

*While Social Determinants of Health (SDoH) are widely acknowledged as critical factors influencing health outcomes, particularly in vulnerable populations, their complex relationships and systemic impacts remain insufficiently examined. This study presents the development and systematic analysis of a comprehensive knowledge graph (KG) framework designed to elucidate the complex relationships between SDoH and mental health outcomes in a high-risk population: veterans with documented histories of suicide attempts or suicidal ideation. Leveraging a comprehensive electronic health records dataset from the U.S. Veterans Health Administration, we generated synthetic data that accurately preserves the statistical properties of the original dataset. We also constructed a specialized SDoH knowledge graph to enable multidimensional analysis. Using topological link prediction and node classification algorithms, we systematically analyzed structural patterns across critical SDoH domains to uncover latent relationships within the KG. Our KG-based approach enables privacy-preserving health disparities research by combining synthetic data generation with graph-based analytics. Our results demonstrate the viability of this approach for deriving clinically meaningful insights while maintaining strict confidentiality protections, establishing a scalable paradigm for future population health studies.*

*Index Terms*—**Knowledge Graph, Mental Health, Suicide, Social Determinants of Health, Veterans, Synthetic Data**

## I. INTRODUCTION

Social Determinants of Health (SDoH) encompass a wide range of non-medical factors that commonly include economic stability, access to quality education, social and community dynamics, neighborhood conditions, and healthcare availability [1]. SDoH factors arise from the complex interplay of social, economic, and environmental forces, directly influencing individuals' ability to achieve optimal health. For example, economic stability enables access to essentials like nutritious food and safe housing, while robust social connections strengthen mental resilience and provide critical support during challenging times [2]. Addressing SDoH is essential to improve public health outcomes, reduce disparities, and design effective interventions [2,3].

SDoH plays a crucial role in mental health, especially among vulnerable populations such as veterans. Disparities in economic stability, social support, and healthcare access contribute to elevated risks of adverse outcomes like suicide or suicidal ideation [4]. Veterans face heightened suicide risk due to factors such as post-traumatic stress disorder (PTSD), depression, substance use disorders, homelessness, and justice system involvement [5]. To address this, the U.S. Department of Veterans Affairs (VA) has implemented prevention initiatives, including the Veterans Crisis Line, predictive modeling for suicide risk, and expanded mental health services [6,7].

Despite these efforts, suicide remains a leading cause of death among veterans [7]. Early identification through predictive models enables timely interventions like crisis counseling and support services [8]. However, many at-risk individuals do not seek help due to stigma, low awareness, or access barriers. Continued research is critical to strengthen prevention strategies, reduce suicide rates, and enhance veteran well-being [9]. Therefore, a deep understanding of the SDoH and health outcome interplay is essential for developing data-driven policies and targeted interventions [10].

Analyzing SDoH data poses challenges due to the sen-

* Equal contribution

sitivity and privacy concerns surrounding personal health information [11]. Synthetic data offers a promising alternative, preserving statistical properties while enhancing privacy. In contrast, traditional anonymization often compromises data utility for privacy. Additionally, conventional analytical methods (e.g., regression) often struggle to capture the complex, non-linear relationships between SDoH factors and health outcomes. In contrast, KGs offer a more adaptable framework by seamlessly integrating multiple data sources, improving their capacity to model these intricate connections.

In this study, we examine the relationship between SDoH and suicide risk patterns in a large national cohort of VA patients in the U.S. using privacy-preserving graph-based methodologies. Our study makes these key contributions.

- We propose a privacy-preserving knowledge graph for SDoH research, enabling secure analysis of complex health data.
- We apply advanced topological link prediction to reveal novel insights, advancing analytical capabilities in this domain.
- We demonstrate how synthetic data can advance healthcare research as a scalable, privacy-preserving alternative to traditional sources.

## II. RELATED WORK

Social determinants of health (SDoH) play a critical role in shaping mental health outcomes such as suicide risk, with economic stability, social cohesion, and healthcare access identified as key drivers [12]. Disruptions in daily routines, reduced social engagement, and altered sleep patterns have also been shown to significantly impact mental health [13]. Housing instability, in particular, has been correlated with elevated suicide risk, emphasizing the need for targeted interventions [14]. These findings underscore the complex connections between SDoH and individual health [13], but often rely on sensitive personal health information, raising substantial privacy challenges [15]. To address these concerns, synthetic data offers a solution to avoid direct links to real individuals, unlike de-identified data. Synthetic datasets are created to preserve the statistical properties and analytical utility of original data while safeguarding individual patient privacy [16]. Studies leveraging synthetic data have demonstrated its potential in training machine learning (ML) models and conducting large-scale analyses without compromising privacy [17]. These advances enable privacy-aware analysis of sensitive health data, though challenges persist in regulatory compliance and risk mitigation [18].

Different ML techniques have been employed to predict suicide risks, offering advanced analytical capabilities for identifying at-risk individuals. For instance, ML algorithms have been developed to predict suicide attempts using longitudinal clinical data from adolescents, achieving high predictive accuracy [19]. Similarly, ML techniques have been applied to Electronic Health Records (EHRs) to predict suicide risk effectively [20]. Another study utilized ML models on large single-payer healthcare registry data (from Denmark) to predict sex-specific suicide risk, demonstrating the potential of ML in personalized risk assessment [21]. These studies underscore the potential of ML in suicide risk prediction, emphasizing the importance of selecting appropriate algorithms and integrating comprehensive data sources to improve predictive accuracy.

KGs offer a principled way to represent structured domain knowledge, enabling semantic consistency and interpretability of complex, multi-relational data [22]. Unlike conventional data mining algorithms that capture statistical associations without explicit relational context, graph representation can provide a robust framework for modeling intricate relationships within multi-dimensional healthcare datasets [23]–[25]. They have been used to enhance suicidal ideation detection through structured analysis of mental health patterns. For instance, one study integrated suicide-oriented personal KGs with deep learning for improved detection on social media [26], while another proposed a KG-based framework for suicide monitoring and crisis intervention, enabling real-time risk assessment [27]. However, the integration of KGs with synthetic data remains under-explored, particularly in SDoH and mental health research.

Our work bridges these gaps by combining synthetic data with KGs for privacy-preserving SDoH analysis. We further introduce topological link prediction to reveal hidden relationships between SDoH and mental health, offering novel insights into associated risk factors [28].

## III. METHOD

### A. Data Source

In this study, we customized the MDClone system [29], to generate synthetic data from the Veterans Health Administration's (VHA) EHR warehouse via the VA Informatics and Computing Infrastructure (VINCI). This privacy-preserving synthetic dataset retains the statistical fidelity of veteran health data [29]. MDClone's platform includes a structured data lake, query engine, and synthetic data generator, producing 784,339 records that preserve the original EHR's multivariate distributions across subpopulations [30].

We characterize the study population using demographic and clinical attributes such as *age*, *gender*, *ethnicity*, *vital signs*, and *comorbidities*. Geographic patterns were assessed using Rural-Urban Commuting Area (RUCA) codes [31], a USDA classification based on commuting flows. Table I compares five random 10% cohorts ($Real_1$–$Real_5$) from the original data, confirming close statistical alignment with the synthetic dataset and validating the robustness of our generation process.

### B. Cohort for Suicide Risk Analysis

We constructed the KG through a systematic pipeline for efficient data integration and analysis. Cohort identification was performed using the MDClone ASK module to facilitate rapid exploration of veteran population data [29]. The initial cohort was defined based on age and encounter type, and subsequently enriched with diagnosis codes, procedures, health factors, and mental health indicators, including PHQ responses [32,33] and ICD-10 codes for suicidal ideation. We analyzed a synthetic

TABLE I
DEMOGRAPHIC AND CLINICAL CHARACTERISTICS ACROSS DATA ORIGINS

| Characteristic | Real$_1$ | Real$_2$ | Real$_3$ | Real$_4$ | Real$_5$ | Synthetic |
|---|---|---|---|---|---|---|
| Age (Mean (SD)) | 53.82 (15.75) | 53.85 (15.80) | 53.91 (15.72) | 53.79 (15.74) | 53.80 (15.75) | 53.84 (15.71) |
| Sex (Male %) | 699,858 (89.2) | 700,313 (89.3) | 700,298 (89.3) | 700,039 (89.3) | 699,945 (89.2) | 700,099 (89.3) |
| **Ethnicity (%)** | | | | | | |
|   Hispanic or Latino | 47,718 (6.7) | 47,746 (6.7) | 48,012 (6.7) | 47,903 (6.7) | 47,626 (6.7) | 48,012 (6.7) |
|   Not Hispanic or Latino | 643,766 (90.1) | 643,530 (90.1) | 643,508 (90.1) | 643,010 (90.1) | 643,627 (90.1) | 643,812 (90.1) |
| **RUCA (%)** | | | | | | |
|   Highly rural | 9,370 (1.2) | 9,543 (1.2) | 9,409 (1.2) | 9,417 (1.2) | 9,401 (1.2) | 9,330 (1.2) |
|   Rural | 250,832 (32.3) | 250,535 (32.2) | 251,099 (32.3) | 250,610 (32.3) | 250,561 (32.3) | 251,061 (32.3) |
|   Urban | 516,749 (66.5) | 516,908 (66.5) | 516,532 (66.5) | 517,039 (66.5) | 516,970 (66.5) | 516,605 (66.5) |
| **Misc. Characteristics (Mean (SD))** | | | | | | |
|   Death age | 78.07 (12.11) | 78.14 (12.08) | 78.08 (12.08) | 78.05 (12.16) | 78.11 (12.10) | 78.11 (12.11) |
|   Diastolic blood pressure | 75.91 (10.65) | 75.91 (10.66) | 75.92 (10.64) | 75.92 (10.67) | 75.90 (10.65) | 75.91 (10.86) |
|   Systolic blood pressure | 129.58 (17.40) | 129.62 (17.44) | 129.62 (17.40) | 129.64 (17.44) | 129.62 (17.44) | 129.62 (17.81) |
|   Hemoglobin | 99.06 (37.61) | 98.98 (37.57) | 98.92 (37.54) | 99.03 (37.51) | 99.01 (37.61) | 99.71 (37.63) |
|   Asthma = Yes (%) | 69,868 (8.9) | 69,574 (8.9) | 69,541 (8.9) | 69,422 (8.9) | 69,657 (8.9) | 69,505 (8.9) |

patient cohort ($n$ = 111,159), aged 18–120, with emergency, inpatient, or primary care encounters between 2010–2022, selected based on documented suicide risk indicators. The study population was defined using validated mental health measures: suicidal ideation via PHQ-9 item 9 [32,33] and the C-SSRS [34], and depression severity via the full PHQ-9 [32]. Clinical data, including ICD-10 codes for self-harm and suicidal behavior history, were integrated to ensure comprehensive risk identification. We queried the MDClone Synthetic Data Lake (SDL) using ICD-10 codes from prior work [35] to extract patient-level SDoH factors. These were categorized into five categories—*Economic Stability*, *Education Access and Quality*, *Healthcare Access and Quality*, *Neighborhood and Built Environment*, and *Social and Community Context*, based on the Healthy People 2030 framework [36]. The processed dataset was exported in structured CSV format to facilitate downstream analytical workflows.

### C. Knowledge Graph Construction

The synthetic dataset was integrated into the BioCypher framework [37], which automated the conversion of structured records into a property graph. Nodes and edges were instantiated to represent entities and relationships across key SDoH domains, with semantic standardization enforced through the BioLink ontology [38]. The resulting graph components were then imported into Neo4j graph database, producing a comprehensive KG.

A partial view of the KG data model (meta-graph) and instance (sub-graph) is shown in Fig. 1. The meta-graph defines the schema centered on the `Patient` node (blue) and its links to demographics (yellow; e.g., gender, ethnicity, race, county), clinical factors (green; e.g., conditions, procedures, surveys), and SDoH factors (pink; e.g., economic stability, education access, social context). Arrows represent relationships such as `HAS_GENDER`, `DIAGNOSED_WITH`, and `RESIDES_IN`, enabling structured integration of heterogeneous health data. The subgraph shows a patient-level instance, connecting a patient to specific demographic, clinical, and SDoH attributes (e.g., "Lack of Housing," "Homeless," and "SUICIDE RISK"), il-lustrating the KG's ability to represent complex, multi-domain health profiles. The final KG comprises 805,625 nodes and 5,942,093 relationships spanning 18 relationship types, with 36 node labels and 60 property keys encoding clinical and SDoH information. This supports granular, patient-centered queries and multilevel analysis of clinical-social interactions, offering distinct advantages for healthcare research.

The final graph comprises 805,625 nodes representing various entities, with 36 distinct node labels defining different types of entities. It contains 60 property keys, meaning that each entity can hold multiple attributes. Property keys are attribute-value pairs assigned to nodes (e.g., patients, conditions, or SDoH factors) or relationships, and they encode metadata such as age, diagnosis codes, geographic location, or survey responses. These fields support rich semantic queries and enable detailed exploration of the graph structure. The total number of relationships connecting these nodes is 5,942,093, distributed across 18 different relationship types. These statistics highlight the complexity and richness of the KG, emphasizing its potential for data analysis, patient insights, and healthcare research.

### D. Topological Link Prediction

We employ topological link prediction to quantify node associations and uncover latent relationships within the KG, leveraging structural proximity, shared neighbors, and connectivity patterns often overlooked by traditional analytical methods. We implement five of the most common link prediction algorithms: *Adamic Adar Index*, *Common Neighbors*, *Preferential Attachment*, *Resource Allocation*, and *Total Neighbors*. These methods assess similarity based on local topology: *Adamic Adar* and *Resource Allocation* weight shared neighbors, *Preferential Attachment* estimates link likelihood from node degree, and the others rely on neighbor overlap and count-based proximity.

The five listed topological link prediction algorithms have been employed to quantify associations between SDoH factors and health indicators (diagnoses, health factors, survey responses), where higher scores reflected stronger relationships.

Fig. 1. The data model and instance model of the KG. A meta-graph in Neo4j represents the schema or high-level structure of the data, defining possible node labels, relationships, and properties. It acts as a template for the actual graph data. An instance model is a concrete example of data stored in Neo4j graph database, a sub-graph representing real entities and relationships. Two example patients (with node instances) are shown on the right.

To ensure robustness, we aggregated algorithm outputs using their geometric mean, reducing individual method biases.

Furthermore, to analyze the true probability of the predicted links, we also assess the statistical significance of our results. We set the null hypothesis ($H_0$) that link prediction scores are not significantly different from random node pairs and the alternative hypothesis ($H_1$) that they are significantly different.

In particular, we tested $H_0$ by creating a randomized graph using a degree-preserving edge-swapping method, maintaining the original graph's structure while randomizing connections. Link prediction scores are computed for both the original and randomized graphs. Using a permutation test, we compare observed scores to the randomized distribution to calculate p-values and run the statistical significance tests.

To address false positives from multiple tests, we apply Bonferroni and Benjamini-Hochberg (BH) corrections. The Bonferroni method controls the family-wise error rate (FWER), while the BH procedure controls the false discovery rate (FDR), making it more suitable for large-scale analyses.

## IV. EXPERIMENTS

### A. Suicidal Risk Prediction - Baseline Models

To assess the predictive value of the KG structure, we established a baseline using tabular feature representations derived from the same patient-level clinical and SDoH features used to construct the KG. We trained three standard machine learning models: XGBoost, Random Forest, and Logistic Regression on these tabular inputs extracted from the synthetic dataset. This setup enables a direct comparison with graph-based models, quantifying the added value of graph embeddings that capture structural relationships within the KG. Performance results for both approaches are reported in Table III.

### B. Suicidal Risk Prediction - Graph Embedding Models

To evaluate suicide risk prediction using KG embeddings, we developed a three-stage pipeline: (1) FastRP was used to generate topological embeddings capturing patient-clinical-SDoH relationships; (2) classifiers (XGBoost, Random Forest, Logistic Regression) were trained on resampled data using SMOTEENN to address class imbalance; and (3) evaluation was performed using a 10% holdout test set, with the remaining data split 80/20 for training and validation. Hyperparameters were tuned via GridSearchCV. To isolate the impact of graph topology, baseline models used identical classifiers and feature sets. While this study focused on FastRP, future work will explore advanced embeddings such as Node2Vec and GraphSAGE.

## V. RESULTS

### A. Associations between SDoH factors and Health Indicators

Table II summarizes the associations found significant between SDoH factors and health indicators across five domains: *Economic Stability*, *Education Access and Quality*, *Healthcare Access and Quality*, *Neighborhood and Built Environment*, and *Social and Community Context*. Each domain includes specific factors linked to health categories, with normalized geometric mean scores, p-values, and Benjamini-Hochberg(BH)-adjusted p-values indicating association strength and significance. Only statistically significant relationships are reported.

The *Economic Stability* domain shows strong associations between housing insecurity (e.g., homelessness, lack of housing) and adverse mental health outcomes (e.g., psychosocial conditions, suicidal ideation), with large effect sizes (0.396–0.927) and high significance (p<0.001; e.g., p=0.00021 for housing status). While unlikely to be random, further research is needed to establish causality and clinical relevance.

TABLE II
Summary of association between SDoH factors and health indicators found to be significant. The higher-level domain for each SDoH factor and the category for each health indicator are shown in the adjacent right column. Norm: normalized geometric mean score, BH Adj: Benjamini-Hochberg (BH).

| SDoH Domain | SDoH Factor | Health Category | Health Indicator | Norm. Score | P-value | BH Adj. |
|---|---|---|---|---|---|---|
| Economic Stability | Lack of Housing | Condition | Unspecified psychosocial circumstance | 0.396 | 0.00021 | 0.04461 |
| | Vitamin D deficiency, unspecified | Health Factor | PTSD SCREEN POSITIVE | 0.927 | 0.00006 | 0.03076 |
| | Homelessness | Health Survey | Over the past month, have you wished you were dead or wished you could go to sleep and not wake up? (answer: Yes) | 0.401 | 0.00304 | 0.04611 |
| Education Access and Quality | Educational circumstances | Condition | Suicidal ideations | 0.134 | 0.00041 | 0.04704 |
| | Problems related to education and literacy, unspecified | Health Factor | PTSD SCREEN POSITIVE | 0.287 | 0.00020 | 0.03832 |
| | Other specified family circumstances | Health Survey | Thoughts that you would be better off dead or of hurting yourself in some way (answer: Several days) | 0.182 | 0.01043 | 0.04487 |
| Healthcare Access and Quality | Residence remote from hospital or other health care facility | Condition | Other psychological or physical stress, not elsewhere classified | 0.428 | 0.00209 | 0.03823 |
| | Unspecified problem related to medical facilities | Health Factor | PTSD SCREEN POSITIVE | 0.553 | 0.00091 | 0.02744 |
| | Other problems related to medical facilities | Health Survey | Over the past month, have you wished you were dead or wished you could go to sleep and not wake up? (answer: Yes) | 0.546 | 0.02912 | 0.04077 |
| Neighborhood and Built Environment | Personal history of military deployment | Condition | Unspecified psychosocial circumstance | 0.703 | 0.00012 | 0.02711 |
| | Exposure to disaster, war and other hostilities | Health Factor | PTSD SCREEN POSITIVE | 0.321 | 0.00037 | 0.03674 |
| | Personal history of return from military deployment | Health Survey | Thoughts that you would be better off dead or of hurting yourself in some way (answer: Several days) | 0.959 | 0.00153 | 0.03387 |
| Social and Community Context | Problem related to unspecified psychosocial circumstances | Condition | Unspecified psychosocial circumstance | 0.628 | 0.00004 | 0.01749 |
| | Other psychological or physical stress | Health Factor | PTSD SCREEN POSITIVE | 0.367 | 0.00016 | 0.02829 |
| | Reaction to severe stress, unspecified | Health Survey | Over the past month, have you wished you were dead or wished you could go to sleep and not wake up? (answer: Yes) | 0.367 | 0.00324 | 0.04289 |

In *Education Access and Quality* domain, educational disadvantages (e.g., literacy challenges, limited access) are significantly associated with suicidal ideation and PTSD, though with modest effect sizes (0.134–0.287). These findings suggest a link between education barriers and mental health, warranting deeper causal investigation.

The *Healthcare Access and Quality* domain shows moderate-to-strong associations (0.428–0.553) between healthcare barriers (e.g., geographic remoteness, facility inadequacies) and mental health outcomes (psychological stress, PTSD), highlighting healthcare access as a protective factor.

The *Neighborhood and Built Environment* domain demonstrates strong associations (0.703–0.959) between trauma exposures (e.g., military deployment, disasters) and mental health conditions (PTSD, psychosocial issues), emphasizing the role of environmental trauma and the need for targeted interventions.

The *Social and Community Context* domain links psychosocial stressors to mental health outcomes, with effect sizes ranging from 0.367–0.628. These findings underscore the importance of social support in mitigating mental health risks.

Overall, these results reveal strong SDoH-mental health associations among veterans, which are aligned with prior studies [4,5], and underscore the need for integrated strategies that address both clinical and social determinants of health.

### B. Suicidal Risk Prediction

Table III compares six models for suicidal risk prediction: three traditional classifiers (XGBoost, Random Forest, Logistic Regression) using tabular features, and three graph-enhanced models with `FastRP` KG embeddings. By controlling for data inputs, this comparison isolates the value of graph-based representations. While tabular models set strong baselines, the graph-based approaches outperform them, demonstrating the predictive advantage of incorporating relational patterns.

Among the baseline models using traditional tabular features, XGBoost achieved the highest accuracy at 0.935, with an F1-score of 0.742 and an AUC-ROC of 0.963. Random Forest performed slightly worse in terms of accuracy and F1-score, but its AUC-ROC was comparable to XGBoost. Logistic Regression had the lowest performance among the three, with an accuracy of 0.915, an F1-score of 0.618, and an AUC-ROC of 0.903. Overall, XGBoost and Random Forest performed similarly well, while Logistic Regression lagged behind in terms of F1-score and AUC-ROC.

TABLE III
PERFORMANCE OF BASELINES VS. KG-BASED (THROUGH FASTRP) MODELS IN BINARY PREDICTION OF SUICIDE RISK

| Data | Model | Accuracy (95% CI) | F1-Score (95% CI) | AUC-ROC (95% CI) |
|---|---|---|---|---|
| Tabular | Random Forest | 0.932 [0.932, 0.933] | 0.721 [0.718, 0.723] | 0.957 [0.956, 0.958] |
| | Logistic Regression | 0.915 [0.913, 0.917] | 0.618 [0.614, 0.622] | 0.903 [0.900, 0.906] |
| | XGBoost | **0.935 [0.935, 0.936]** | **0.742 [0.740, 0.744]** | **0.963 [0.962, 0.964]** |
| KG Embedding (FastRP) | Random Forest | 0.960 [0.959, 0.961] | 0.850 [0.847, 0.853] | 0.990 [0.989, 0.991] |
| | Logistic Regression | 0.965 [0.963, 0.966] | 0.885 [0.882, 0.889] | 0.993 [0.992, 0.994] |
| | XGBoost | **0.983 [0.982, 0.983]** | **0.937 [0.934, 0.939]** | **0.996 [0.996, 0.997]** |

In suicide risk prediction, graph-based models using FastRP embeddings consistently outperform tabular baselines across all metrics. XGBoost, in particular, achieves the highest overall performance with 0.983 accuracy, 0.937 F1-score, and 0.996 AUC-ROC, making it the most suitable candidate for deployment. Logistic Regression and Random Forest also show strong performance in the graph-based setup. These results highlight the value of incorporating relational structures such as SDoH interdependencies into the pipeline, motivating further exploration of graph-based approaches.

## VI. DISCUSSION

This study explores the intricate relationships between social determinants of health (SDoH) and mental health outcomes among veterans using synthetic data and knowledge graph (KG) methodologies. We constructed a privacy-preserving KG with over 800,000 nodes and 5.9 million relationships, enabling scalable and interpretable modeling of interdependencies across five key SDoH domains: *Economic Stability*, *Education Access and Quality*, *Healthcare Access and Quality*, *Neighborhood and Built Environment*, and *Social and Community Context*. Using topological link prediction algorithms, we identified latent associations that conventional models (e.g., logistic regression) often miss. Our findings indicate that housing stability reduces stress and suicidal ideation risk; educational barriers elevate PTSD and suicidal thoughts; healthcare access issues, especially in remote areas, worsen psychological distress; and military-related environmental exposures increase mental health vulnerabilities. Social and community connections emerged as protective factors, highlighting the need for support-driven interventions.

Overall, this work demonstrates the utility of KG-based approaches in uncovering structural drivers of health inequities and offers a data-driven foundation for targeted policy and intervention design. A key strength of this study lies in its methodological integration of synthetic data and knowledge graphs (KGs), enabling large-scale, privacy-preserving analyses of sensitive health information. This framework ensures data confidentiality while supporting scalable and reproducible investigation of complex health phenomena. While synthetic data may differ from real-world distributions, prior work has shown that MDClone-generated data preserves complex multivariate dependencies and population-level characteristics [30]. Our own validation (Table I) confirms high concordance in demographic and clinical attributes, supporting its utility for exploratory analysis. Although individual-level fidelity cannot yet be guaranteed, we are awaiting approval to validate our models on real-world VHA patient data within a privacy-compliant framework.

As part of our ongoing work, we also plan to explore additional graph embedding techniques (e.g., Node2Vec, Graph-SAGE) and evaluate advanced classifiers and LLMs to further enhance predictive performance and generalizability. However, individual-level fidelity is not guaranteed, and future work is needed to replicate these findings in real patient data. We are actively pursuing privacy-compliant environments to enable such validation. Although the observational design limits causal inference, this work establishes a secure and generalizable foundation for advancing research into the interplay between SDoH and mental health.

## VII. CONCLUSION

This work demonstrates the potential of synthetic data and knowledge graphs (KGs) to advance health informatics and promote equitable, data-driven care, particularly for vulnerable populations. By integrating privacy-preserving KG methods with link prediction algorithms, we developed a scalable framework to analyze the complex relationship between social determinants of health (SDoH) and mental health outcomes among veterans. This approach enhances insight discovery and lays the groundwork for future mental health research. Our findings deepen the understanding of how SDoH impact mental health and offer guidance for targeted interventions and policy development to reduce health disparities. The use of synthetic data addresses privacy concerns, enabling wider application in healthcare. Future work will focus on validating results with real-world data and incorporating additional sources, such as unstructured clinical notes, to further refine the framework and support actionable insights for health systems and policymakers.

### ACKNOWLEDGMENTS

This work was supported by the NSF awards 2333740 and 2443639, as well as NIH award P20GM103446. This material is the result of work supported with resources at Wilmington VA Medical Center, Wilmington, Delaware. The content does not represent the views of the United States Department of Veterans Affairs or the United States Government.

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
