# OpenReview forum: "Leveraging Social Determinants of Health (SDoH) Knowledge Graph to Identify Latent Patterns in Veteran Suicide Risk"
_IEEE.org/EMBS/BHI/2025/Conference — BHI 2025_

### Official Review · Reviewer_9JhZ · 2025-07-12
**Leveraging Social Determinants of Health (SDoH) Knowledge Graph to Identify Latent Patterns in Veteran Suicide Risk**

**Confidence:** 5
**Clarity Of Writing:** great
**Clinical Significance:** great
**Methodological Novelty:** great
**Overall Rating:** 8

**Experiments And Results:**

great

**Questions For The Authors:**

How does the model performance vary when incorporating different levels of granularity in SDoH features—such as specific education levels or housing durations—rather than broad categorical groupings?

Have you considered applying temporal knowledge graph techniques to capture longitudinal patterns in SDoH changes over time, and if so, what challenges do you anticipate in adapting your current framework for such dynamic modeling?

**Strengths:**

The paper’s greatest strengths lie in its innovative integration of synthetic data generation with a social determinants of health knowledge graph, providing a privacy-preserving yet richly relational dataset. Its use of graph-based link prediction and embedding techniques effectively uncovers latent associations between social factors and veteran suicide risk. Moreover, it demonstrates improved predictive performance by combining relational graph features with traditional machine learning models, offering a powerful and scalable framework for complex mental health analysis.

**Summary Of The Paper:**

The paper presents a privacy-preserving framework that combines synthetic electronic health records and knowledge graphs to analyze the relationship between social determinants of health and suicide risk among U.S. veterans. Using topological link prediction and graph embedding techniques, the authors uncover significant associations between factors such as housing instability, education, and healthcare access with mental health outcomes like PTSD and suicidal ideation. Graph-based models outperform traditional machine learning models in suicide risk prediction, highlighting the added value of relational data representation in capturing complex social-clinical interactions.

**Weaknesses:**

The study relies entirely on synthetic data, which, despite preserving population-level distributions, may not capture individual-level nuances necessary for certain types of clinical inference. Additionally, the analysis focuses on FastRP for graph embeddings without exploring other modern techniques like Node2Vec or GraphSAGE, which may offer complementary or superior performance.

---

### Official Review · Reviewer_CRPG · 2025-07-13
**This paper builds a large-scale knowledge graph (KG) using synthetic EHR data from the US Veterans Health Administration to study relationships between social determinants of health (SDoH) and suicide risk.**

**Confidence:** 3
**Clarity Of Writing:** good
**Clinical Significance:** great
**Methodological Novelty:** fair
**Overall Rating:** 4

**Experiments And Results:**

good

**Questions For The Authors:**

1. Could you detail the mapping process between structured data (ICD-10, survey responses) and SDoH domains? Was manual review or external labeling used?
2. Did you consider modeling with modern GNNs? Why did you choose FastRP over more expressive methods such as GraphSAGE, GAT, or Node2Vec, especially since your KG includes node attributes?
3. Are SDoH concepts like “housing” and “homelessness” modeled with semantic hierarchy or treated as distinct flat nodes?
4. How did you deal with sparsity or disconnected nodes in the KG? Were there many singletons (nodes with no neighbors)? How do you treat them during embedding?
5. What was the suicide label source? How was class imbalance handled before/after SMOTEENN?'
6. Given that the entire study is based on synthetic data, how do you plan to validate these findings in a real-world clinical cohort? Could your graph-based model generalize to non-veteran datasets?
7. Did you attempt any form of causal modeling or downstream simulation to evaluate whether interventions on high-impact SDoH nodes (e.g., housing) would affect suicide risk outcomes?
8. Are all edges treated as unweighted and undirected? Were different types of SDoH–outcome links (e.g., direct clinical, inferred social) encoded differently?

**Strengths:**

1. Veteran suicide is a well-documented national crisis. SDoH offer actionable, policy-relevant insights often overlooked in standard clinical prediction models.
2. The graph construction using BioCypher + Neo4j with Biolink-model alignment shows reasonable implementation and good use of standard biomedical ontologies. The final KG includes over 800,000 nodes and ~6 million edges, enabling scalable graph learning.
3. Using MDClone synthetic data ensures privacy while retaining meaningful structure.
4. Incorporation of five link prediction algorithms + ensemble ranking (geometric mean + permutation testing) adds robustness.
5. Suicide risk classification improves significantly when adding FastRP embeddings: F1 rises from 0.61 to 0.88 for logistic regression.

**Summary Of The Paper:**

The authors extract over 110,000 patient records, use BioCypher and the Biolink model for semantic structuring, and construct a Neo4j graph with over 800,000 nodes. They apply classical topological link prediction methods (e.g., Adamic-Adar, Resource Allocation) to identify latent associations, and embed the graph using FastRP to improve suicide risk prediction, outperforming standard ML models (e.g., logistic regression, random forest).

**Weaknesses:**

1. While the implementation is solid, the core novelty is relatively modest. The use of topological link prediction on KGs and FastRP embedding is not new. There is limited methodological development beyond standard graph mining and ML.
2.  The process of extracting and encoding SDoH relationships is not fully explained. For example:
	How are ambiguous ICD-10 codes mapped to specific SDoH domains? Are there any NLP or manual validation steps involved?
	How are nodes with overlapping semantics resolved in the ontology (e.g., “homelessness” under both SDoH and clinical)?
3. The results are based entirely on synthetic data and do not involve any external dataset or real-world confirmation. While synthetic data preserves statistical properties, causal discovery or policy translation should be cautious.
4. Only three traditional ML models are used as baselines. No comparisons to modern deep learning-based graph methods (e.g., GCN, GAT) are provided.
5. While the authors report some associations (e.g., housing → PTSD), the clinical interpretability of FastRP embeddings or link prediction outcomes is not discussed. How would these insights be used in real VA systems and also any biological mechanisms linked to these findings?
6. Several associations like “Homelessness → suicidal ideation” are already known and well-documented in many papers. It is unclear what genuinely new insight the this KG added over tabular statistical analysis.

---

### Official Review · Reviewer_H3af · 2025-07-17
**Leveraging Social Determinants of Health (SDoH) Knowledge Graph to Identify Latent Patterns in Veteran Suicide Risk**

**Confidence:** 4
**Clarity Of Writing:** good
**Clinical Significance:** good
**Methodological Novelty:** good
**Overall Rating:** 7

**Experiments And Results:**

good

**Questions For The Authors:**

No question

**Strengths:**

This paper proposes a privacy-preserving knowledge graph for SDoH research, enabling secure analysis of complex health data, and demonstrates how synthetic data can advance healthcare research as a scalable, privacy-preserving alternative to traditional sources.

**Summary Of The Paper:**

This paper presents the development and systematic analysis of a comprehensive knowledge graph (KG) framework aimed at uncovering the complex relationships between social determinants of health (SDoH) and mental health outcomes in a high-risk population—veterans with documented histories of suicide attempts or suicidal ideation.

**Weaknesses:**

No question

---

### Official Review · Reviewer_UPZe · 2025-07-18
**Predicting veteran suicidal risk on synthetic data with SDoH Knowledge Graph**

**Confidence:** 4
**Clarity Of Writing:** good
**Clinical Significance:** great
**Methodological Novelty:** fair
**Overall Rating:** 2

**Experiments And Results:**

poor

**Questions For The Authors:**

1. The resolution in Figure 1 is low, it is hard to read the word in the figure. Please improve the resolution.
2. Why is the KG graph performs worse with Xgboost model? Typically, Xgboost model is one of the best model for classification task in tabular data.
3. Provide a list of variables SDoH factors and health indicators you used as candidate predictors.
4. Please perform the validation of your model on the real VHA dataset to confirm the performance.
5. Please provide the confidence interval for the AUROC and other metrics.
6. Examine more embedding techniques of graphs and additional ML algorithms to demonstrate the statement that KG graph can improve the predictive performance for suicidal risk.

**Strengths:**

1. Veteran suicide prevention is a critical public health issue. This paper’s focus on leveraging SDoH data is highly relevant and impactful.

**Summary Of The Paper:**

This paper presents a synthetic Electronic Health Records (EHRs) from the Veterans Health Administration with a custom-built Social Determinants of Health (SDoH) knowledge graph (KG) to study suicide risk among veterans. The study constructs a knowledge graph using the BioCypher and Neo4j platforms and applies topological link prediction and graph embedding methods (FastRP) to identify associations and improve prediction of suicide risk. The authors benchmark performance against 3 tabular ML (xgboost, logistic regression and random forest) models. I think this is an important clinical research area and interesting strategy to use knowledge graph for developing predictive analytics. However, I am not convinced either 1) KG improve the performance or 2) the 99% AUC on the synthetic data really represent the results for real veterans suicidal risks.

**Weaknesses:**

1. The performance of the suicidal prediction is very good even without the KG graph (Xgboost AUROC = 0.96). However, as I have been searching for state-of-the-art performance in real VHA dataset, the performance is 80 - 85% AUROC (a). PMID: 32435212, b) Mitra, Avijit, et al. "Predicting Suicide Among US Veterans Using Natural Language Processing-enriched Social and Behavioral Determinants of Health." Research Square (2024): rs-3.) This raise my concern of the performance inflation due to the use of synthetic data. I would strong suggest to validate your model on the real VHA data, it could be a small subset for validation.

2. The enhanced ML models using knowledge graph (KG) embeddings do not consistently outperform the baseline tabular models. In fact, one out of the three models—XGBoost—performs worse with KG embeddings. This limited improvement is insufficient to conclusively demonstrate the superiority of the KG-based approach. Additional analysis is needed, including the use of more diverse embedding techniques (e.g., Node2Vec, GraphSAGE) and a broader range of machine learning algorithms to strengthen the claim.

---

### Official Review · Reviewer_qt1g · 2025-07-18
**Knowledge graph for suicide risk predictions**

**Confidence:** 3
**Clarity Of Writing:** good
**Clinical Significance:** great
**Methodological Novelty:** fair
**Overall Rating:** 7

**Experiments And Results:**

good

**Questions For The Authors:**

Can the methodology applicable to a general population?

**Strengths:**

- good synthetic dataset construction that enables privacy-aware analysis on sensitive health data
- strong performance demonstrated using graph embeddings for suicide risk classification
- address a high-impact problem that have positive social and clinical outcomes

**Summary Of The Paper:**

This paper develops a knowledge graph (KG) framework designed to predict suicide risk in veterans. The relationship between Social Determinants of Health (SDoH) and suicide risk patterns is identified using a privacy preserved method by constructing a synthetic dataset from EHR from US Veterans Health Administration. The synthetic data is analyzed using a knowledge graph that enables high dimensional analysis revealing the strong associations between multiple SDoH factors - particularly housing instability, education barriers, and healthcare access, as well as mental health outcomes like PTSD and suicidal ideation.

**Weaknesses:**

- please check your paper format to be compliant with IEEE format
- lacks validation of external data as well as benchmark with existing state-of-the-art method (e.g., GraphSAGE)